# Caffeic Acid Phenethyl Ester Alleviates Alcohol-Induced Inflammation Associated with Pancreatic Secretion and Gut Microbiota in Zebrafish

**DOI:** 10.3390/biom15070918

**Published:** 2025-06-22

**Authors:** Menghui Lin, Xiaogang Guo, Xinyu Xu, Chao Chang, Thanh Ninh Le, Haiying Cai, Minjie Zhao

**Affiliations:** 1School of Biological and Chemical Engineering, Zhejiang University of Science & Technology, Hangzhou 310023, China; 212403817025@zust.edu.cn (M.L.); 13709097644@zust.edu.cn (X.G.); 1230410042@zust.edu.cn (X.X.); l1451259903@gmail.com (C.C.); 2College of Biosystems Engineering and Food Science, Zhejiang University, Hangzhou 310058, China; lethanhninh@tuaf.edu.vn; 3Institute of Biotechnology and Food Technology, Thai Nguyen University of Agriculture and Forestry, Thai Nguyen 23000, Vietnam

**Keywords:** caffeic acid phenethyl ester, inflammation, pancreatic secretion, intestinal microbiota, transcriptomics

## Abstract

Caffeic acid phenethyl ester (CAPE) is identified to be an efficacious bioactive polyphenol in propolis for ameliorating glucose and lipid metabolism disorders and inflammation. In this study, an alcohol-induced zebrafish inflammation model was established. CAPE treatments at different concentrations (0.04, 0.2, and 1.0 μg/mL) were administered to alcohol-exposed zebrafish to investigate the underlying mechanisms of alleviating alcohol-induced liver inflammation using transcriptomic analysis and 16S rRNA gene sequencing methods. The results indicated that CAPE decreased the expressions of TNF-α and IL-1β and significantly increased the expression of IL-10 (*p* < 0.0001). Based on the KEGG enrichment analysis of transcriptomic sequencing, CAPE effectively alleviated the inflammation in zebrafish mainly through pancreatic secretion, complement and coagulation cascades, and protein digestion and absorption. Molecular docking supported the potential of CAPE in targeting cholecystokinin (CCK) A Receptor (CCKAR) and mediating the regulation of pancreatic secretion and related inflammation pathways. Moreover, intestinal microbiota analysis demonstrated that CAPE could improve the alcohol-induced microbiota disorder. Additionally, there was a significant correlation between the key genes related to lipid and sterol metabolism among the KEGG-enriched pathways and the specific intestinal microbial communities in zebrafish. Flavobacterium from Bacteroidota was significantly positively correlated with *CEL1*, *CEL2*, and *LPIN* (*p* < 0.01), which suggested that the anti-inflammatory function of CAPE was closely associated with the intestinal microbiota improvement. In conclusion, our findings demonstrated that CAPE could ameliorate liver inflammation in alcohol-induced zebrafish, which was mainly associated with the regulation of pancreatic secretion and intestinal microbiota disorder. This study emphasized the anti-inflammatory mechanisms of CAPE based on targeting the pancreatic secretion pathway, which will broaden the application of natural antioxidants in improving metabolic and inflammatory problems.

## 1. Introduction

It is estimated that nearly one-third of the Chinese population is affected by some form of liver disease, including viral hepatitis, alcoholic liver disease (ALD), non-alcoholic fatty liver disease (NAFLD), liver cirrhosis, and hepatocellular carcinoma (HCC) [1]. Based on data from the World Health Organization (WHO), alcohol consumption accounted for over 3 million deaths in 2016, constituting 5.3% of global mortality. Multiple studies have demonstrated that the excessive accumulation of by-products from alcohol metabolism can trigger oxidative stress, lipid peroxidation, inflammation, and fat deposition [2]. A recent study has shown that alcohol exposure induces oxidative stress and suppresses the expression of hepcidin in the liver [3]. This suppression of hepcidin expression leads to iron overload and subsequently triggers ferroptosis in hepatocytes, thereby exacerbating liver injury. It is believed that early effective intervention is import for impeding the progression of alcohol-induced inflammation and liver disease [4].

A zebrafish (*Danio rerio*) model for alcohol-induced liver inflammation research has several advantages. Firstly, the zebrafish genome exhibits a high degree of homology with the human genome, exceeding 87% [3]. Zebrafish are characterized by their rapid embryonic development. In contrast to rodents, continuous alcohol exposure of zebrafish in the water suffices to induce liver damage and establish an alcohol-induced inflammation model [4]. It is reported that the continuous induction of zebrafish with 0.5% ethanol in feeding water for a period of 2–4 weeks or 2% ethanol for merely 24 h is sufficient to induce a series of ALD symptoms, ranging from steatosis to fibrosis and cirrhosis [5]. Hu et al. found that alcohol could induce liver steatosis in zebrafish, mainly associated with the alteration in the triglyceride metabolism and the PPAR pathway, and the antioxidant N-acetylcysteine (NAC) can effectively inhibit liver steatosis in zebrafish [6]. Dietary supplementation of antioxidants is found to be an effective strategy in improving alcohol-induced inflammation.

Propolis represents a viscoelastic substance that bees gather from various plants. Over 300 chemical constituents, including flavonoids, terpenoids, and phenols, have been identified within propolis [7]. Caffeic acid phenethyl ester (CAPE), a polyphenolic compound that is abundant in propolis, has been identified to be an efficacious bioactive compound in propolis for ameliorating glucose and lipid metabolism disorders [8]. CAPE also plays a role in antimicrobial activity, anti-apoptosis, neuroprotection, analgesia, immunomodulation, cancer prevention, and the prevention of cardiovascular diseases [9]. CAPE manifests antioxidant activity due to its caffeic acid structure part, which is reported to enhance the antioxidant capacity of cells through increasing the activities of superoxide dismutase (SOD), catalase (CAT), glutathione peroxidase (GPx), and effectively scavenging superoxide anions and hydroxyl radicals induced by type III collagen fragments [10]. Furthermore, CAPE can inhibit the lipogenesis via regulating the phosphorylation of ERK and AKT, exerting a weight-loss effect in vivo [11,12]. Our previous study has revealed that CAPE can ameliorate the lipid metabolism disorder in HepG2 cells induced by oleic acid [13], mainly through regulating the oxidation stress and the PPARα-regulated fatty acid oxidation pathway. Nie et al. found that CAPE activated the phosphorylation of IRS-1 and Akt proteins, and simultaneously inhibited the phosphorylation and nuclear translocation of JNK and NK-κB p65 proteins in a type 2 diabetes mellitus (T2DM) model of C57BL/6J mouse [14]. This finding elucidated that CAPE ameliorates T2DM in mice and alleviates insulin resistance by inhibiting the JNK-related inflammatory signaling pathways. Although CAPE exhibits favorable anti-inflammatory and antibacterial properties, the efficiency and precise regulatory mechanism of CAPE in alcohol-induced inflammation remain obscure.

A previous study has indicated that ethanol intake can lead to an instantaneous increase in pancreatic amylase output and plasma cholecystokinin (CCK) levels [15]. CCK was identified as a peptide hormone, which influences diverse processes of nutrient digestion, such as stimulating the secretion of pancreatic amylase, augmenting gallbladder contraction to release bile, and delaying gastric emptying to prolong gastric digestion [16]. The combined action of ethanol and CCK-8 can facilitate the activation of NF-κB in the pancreas, resulting in an up-regulation of pro-inflammatory cytokines and chemokines [17]. Ethanol may enhance the effects of CCK through directly augmenting the sensitivity of acinar cells, and may stimulate the release of CCK from duodenal I-cells [18,19]. The CCK A receptor (CCKAR) plays key roles in the downstream pathways of CCK signaling in many tissues [15], which might be an important potential regulatory target in improving alcohol-induced inflammation and disease.

The gut microbiota serves as the initial barrier and is intricately associated with the inflammatory response and metabolism disorder. Abundant studies have indicated that long-term exposure to special diets can cause dysfunctions in the gut microbiota and gut barriers [20,21]. In this study, an alcohol-induced zebrafish inflammation model was established by exposing the zebrafish to 0.75% ethanol. Transcriptomics and 16S rRNA sequencing were used to comprehensively explore the mechanisms of CAPE in the alcohol-induced inflammatory zebrafish. We also utilized molecular docking to identify potential targets and mechanisms of CAPE in improving inflammation through the pancreatic secretion mediated by cholecystokinin (CCK) A receptor (CCKAR) regulation. This study would provide a reference for applying natural antioxidants in improving alcohol-induced inflammation and other health problems.

## 2. Materials and Methods

### 2.1. Zebrafish Feeding and Grouping Experiment

In the current study, all the procedures for zebrafish experiments complied with provisions on administration of the Laboratory Animals–General Requirements for Animal Experiments (GB/T 35823-2018, China) [22]. All protocols and designs were granted approval from the Institutional Animal Care and Use Committee of the Zhejiang University of Science and Technology (Hangzhou, China).

Caffeic acid phenethyl ester (CAPE, purity ≥ 97%, HPLC verified; CAS No. 104594-70-9, batch C102139) was purchased from Aladdin Biochemical Technology Co., Ltd. (Shanghai, China). Male zebrafish at 4 months post fertilization were purchased from Shanxi Xiyue Biotechnology Co., Ltd. (Shanxi, China), and maintained at 28 °C under a 14 h light/10 h dark cycle, with a density of approximately 10 fish per 1 L of water. These zebrafish were fed a diet (AP105, Feixi biotech Ltd., Shanghai, China) twice a day. The zebrafish were acclimated for one week before the treatment. Then, the experimental fish were distributed among 18 recirculating aquaculture systems and assigned to six experimental diet groups. Each diet was administered to three tanks, each housing 15 fish. Six groups (*n* = 45 for each group) were treated as follows: normal fish-rearing water (negative control group), fish-rearing water supplemented with 0.75% ethanol (EtOH group), fish-rearing water containing 10 mg/L DEX and 0.75% ethanol (positive control group), and 0.75% ethanol solution with CAPE at 0.04 μg/mL, 0.2 μg/mL, and 1 μg/mL concentrations (CAPE0.04, CAPE0.2, and CAPE1 group, respectively). The treatment of CAPE on the experimental fish lasted for 7 days.

### 2.2. Sample Collection and Preservation and Biochemical Analysis of Serum Inflammation Factors

After 7 days of different treatments, 18 zebrafish per group were randomly selected for liver collection (livers of 6 fish pooled as one sample, three parallel samples per group), and 18 adult zebrafish per group for intestinal tract collection (intestinal tracts of 3 fish combined as one sample, six parallel samples per group). All samples were flash-frozen in liquid nitrogen for 30 min and stored in a −80 °C freezer for subsequent transcriptome and intestinal microbiota analysis. Liver tissues were collected from zebrafish in each treatment group, prepared into homogenates, and centrifuged. The supernatants were then collected, and the serum levels of Tumor necrosis factor-α (TNF-α), Interleukin-1β (IL-1β), and Interleukin-10 (IL-10) were measured according to the instructions of the commercial ELISA kits (eBioscience, San Diego, CA, USA).

### 2.3. Transcriptomic Analysis of Zebrafish Liver Sample

mRNA was isolated using the MJZol kit (Shanghai Majorbio Bio-pharm Technology Co., Ltd., Shanghai, China). cDNA was synthesized with random hexamer primers, followed by end-repair, phosphorylation, and adapter addition to the synthesized cDNA. The zebrafish RNA-seq transcriptome libraries were prepared using the Illumina^®^ Stranded mRNA Prep, Ligation kit (San Diego, CA, USA). Magnetic beads were used to select cDNA target fragments sized 300–400 bp, which were then PCR-amplified for 15 cycles. After quantification with Qubit 4.0, the sequencing library was sequenced on the NovaSeq X Plus platform (PE150) with the NovaSeq kit.

The bioinformatics analysis of transcriptomic expression was conducted on the Majorbio Cloud Platform (https://cloud.majorbio.com accessed on 23 January 2024) [23]. For transcriptome bioinformatics analysis, fastp 0.19.5 (https://github.com/OpenGene/fastp, accessed on 23 January 2024) was used to perform quality control on the raw sequencing data. HiSat2 2.1.0 (http://ccb.jhu.edu/software/hisat2/index.shtml, accessed on 23 January 2024) was applied for alignment with the reference genome, followed by the quantification of gene and transcript expression levels using RSEM 1.3.3 (http://deweylab.github.io/RSEM/, accessed on 23 January 2024). Differential expression analysis was carried out using DESeq2 1.24.0 (http://bioconductor.org/packages/stats/bioc/DESeq2/, accessed on 23 January 2024). The Gene Ontology (GO) (http://geneontology.org/, accessed on 23 January 2024) and Kyoto Encyclopedia of Genes and Genomes (KEGG) (https://www.genome.jp/kegg/, accessed on 23 January 2024) databases were used for the functional annotation analysis of differentially expressed genes. Finally, GO enrichment analysis and KEGG pathway enrichment analysis were performed using Goatools (https://github.com/tanghaibao/GOatools, accessed on 23 January 2024) and the Python scipy software package (https://scipy.org/install/, accessed on 23 January 2024).

### 2.4. Intestinal Microbial DNA Extraction and 16S rRNA Sequencing

Total microbial genomic DNA was extracted from colon contents collected after sacrifice using the QIAamp DNA Stool Mini Kit (Qiagen, Hilden, Germany). The V3-V4 hypervariable regions of the bacterial 16S rRNA gene were amplified by PCR on a T100 Thermal Cycler PCR machine (Bio-Rad Laboratories, Inc., Hercules, CA, USA) using the primer set 338F (5′-ACTCCTACGGGAGGCAGCAG-3′) and 806R (5′-GGACTACHVGGGTWTCTAAT-3′) [23]. The PCR products were excised from a 2% agarose gel and then purified using a PCR Clean-up Kit (Shanghai Yuhua Life Science & Technology Development Co., Ltd., Shanghai, China). The purified products were quantified using a Qubit 4.0 (Thermo Fisher Scientific, Inc., Waltham, MA, USA). The purified amplicons were subjected to sequencing on the Illumina Nextseq2000 platform (Illumina, Inc., San Diego, CA, USA).

The bioinformatics analysis of intestinal microbiota was conducted on the Majorbio Cloud Platform (https://cloud.majorbio.com). To create operational taxonomic units (OTUs) of microbes, the 16S rRNA gene sequences were clustered using UPARSE 7.0 with a similarity of greater than 97%. Specifically, Mothur v1.30.1 was used to calculate rarefaction curves and α-diversity indices of intestinal microbes. Based on Bray–Curtis dissimilarity, the Vegan v2.5-3 software package was employed for principal coordinates analysis (PCoA) to determine the similarity of microbial communities among different groups. Linear discriminant analysis (LDA) effect size (LEfSe) (http://huttenhower.sph.harvard.edu/LEfSe, accessed on 26 January 2024) was utilized to identify significantly abundant bacterial taxa.

### 2.5. Molecular Docking for CCKAR

The PDB ID of CCKAR is 7F8U. The structure of CAPE was drawn using ChemDraw 20.0 and then converted into a PDB structure. For the receptor and ligand pre-processing and the docking process, Autodock Tool 1.5.7 was employed. The docking process referred to the previous study. The docking results were visualized using PyMOL 3.1.3. The docking process was performed according to the previous study [24]. In brief, the receptor structure was preprocessed by removing water molecules and isolating the target chain using AutoDockTools (v1.5.7). Non-polar hydrogens were merged into heavy atoms, and polar hydrogens were added. Ligands were preprocessed on the properties of hydrogens, Gasteiger charges, and defined rotatable bonds. The molecular docking was performed using a rigid receptor-flexible ligand model, and the interaction results were visualized in PyMOL 3.1.3.

### 2.6. Data Analysis

The data was expressed as the mean ± standard error (SD). The statistical analysis was conducted using GraphPad Prism 9.0 (Dotmatics, Boston, MA, USA). The significant differences were determined using a one-way analysis of variance (ANOVA) followed by Tukey’s multiple comparisons test. R 2.15.3 (Posit, PBC, Boston, MA, USA) was utilized for the Spearman correlation study.

## 3. Results and Analysis

### 3.1. CAPE Ameliorates Alcohol-Induced Liver Inflammation in Zebrafish

In this study, alcohol-induced zebrafish were treated with 0.04 μg/mL, 0.2 μg/mL, and 1.0 μg/mL of CAPE (CAPE0.04, CAPE0.2, and CAPE1 groups) to explore their regulatory effect on inflammation and related metabolic pathways. According to the serum inflammation factors detection, alcohol treatment remarkably increased the levels of TNF-α and IL-1β in zebrafish livers (*p* < 0.0001 and *p* < 0.0001, respectively), and significantly reduced the level of IL-10 in the liver (*p* < 0.0001) (Figure 1). Compared to the alcohol-induced group (EtOH group), CAPE at various concentrations significantly decreased the serum levels of TNF-α and IL-1β in zebrafish livers (*p* < 0.0001 and *p* < 0.0001, respectively), and significantly increased the serum levels of IL-10 (*p* < 0.0001). Moreover, the regulatory effects of CAPE on these inflammatory factors showed a dosage-dependent manner. IL-10 exerts its effect by inhibiting the activation of signaling pathways involved in immune cell activation and cytokine production [25]. TNF-α and IL-1β are pro-inflammatory cytokines. The increased extracellular IL-1β triggers a phosphorylation cascade that activates transcription factors such as NF-κB and c-Jun, and then translocates them to the nucleus, which initiates the transcription of target genes involved in the inflammatory cascade reaction [26]. This result suggested significant ameliorative effects of CAPE on alcohol-induced inflammation in zebrafish (Figure 1).

### 3.2. Effects of CAPE on Inflammation and Related Pathways in the Liver of Zebrafish Based on Transcriptomic Analysis

Effects of CAPE on inflammation and related pathways in the liver of zebrafish were further investigated using 1.0 μg/mL CAPE treatment samples (the CAPE group) with the control, and the EtOH group based on transcriptomic analysis. According to the transcriptomic analysis results, a total of 15,590 genes of zebrafish were identified. The analysis of differentially expressed genes (DEGs) showed that there were 1892 DEGs between the control group and the EtOH group, while there were 2152 DEGs between the CAPE group and the EtOH group (Figure 2a,b). This indicated that CAPE treatment significantly influenced the gene expression in the liver of alcohol-induced zebrafishes. Furthermore, the principal component analysis (PCA) based on transcriptomic analysis was performed to confirm the significant effect of CAPE (Figure 2b). Samples from the CAPE group were nearly separated with those from the EtOH group, and PC1 and PC2 accounted for 31.43% and 18.91%, respectively, indicating that CAPE treatment strongly influenced the gene expression in the inflammatory liver of zebrafishes.

As the gene expression changes reflect the related pathways of inflammation development, the functional annotation and pathway enrichment analysis on these differential genes were conducted (Figure 2d,e). According to the GO (Gene Ontology) functional annotation, the altered BP (Biological Process) between the CAPE groups and the EtOH group mainly encompassed cellular process (1181 terms), biological regulation (858 terms), and developmental process (719 terms). In addition, the altered CC (Cellular Component) mainly contained cell parts (1346 terms), organelle (754 terms), and membrane parts (695 terms). And the altered MF (Molecular Function) mainly contained binding (1302 terms), catalytic activity (719 terms), and transporter activity (140 terms). Similarly, GO enrichment analysis showed that catalytic activity, ion binding, different metabolic process, and membrane system-related pathways were enriched (Figure 2d). Furthermore, KEGG annotation showed that the effect of CAPE on the hepatic transcriptomes of alcohol-induced zebrafish involved lipid metabolism (65 terms), folding, sorting and degradation (63 terms), signal transduction (223 terms), transport and catabolism (117 terms), the immune system (140 terms), and cancer (166 terms). Consistently, KEGG enrichment analysis based on the comparison of the transcriptomic expression of the CAPE groups and the EtOH group revealed that pancreatic secretion (*p*_adjust_ < 0.001), complement and coagulation cascades (*p*_adjust_ < 0.001), protein digestion and absorption (*p*_adjust_ < 0.01), steroid biosynthesis, protein processing in endoplasmic reticulum, and fat digestion and absorption were significantly enriched (Figure 2e).

According to the KEGG enrichment analysis, the pancreatic secretion was the most prominent pathway (Figure 2). Protein digestion and absorption, bile acid secretion, and fat digestion and absorption were tightly related to pancreatic secretion, indicating that pancreatic secretion might be an important potential target pathway of CAPE in alcohol-induced inflammatory zebrafishes, which would be further substantiated. Further analysis in the gene expression of multiple pancreatic digestive enzymes in pancreatic secretion showed significant alterations in the CAPE group compared to the EtOH group, with a significantly downward trend in the expression of pancreatic amylases, pancreatic lipases, and pancreatic proteases (Figure 3a). The expression of key genes in inflammation and the related pathways were also identified to be significantly changed (Table 1). Compared to the control group, the key proteins in complement and coagulation cascades pathway, including complement proteins C3, C5, C6, C7, C8, and C9 and coagulation protein F2, F5, F8, F9, and F10, were significantly upregulatedly expressed in the EtOH group (Table 1). And this tendency was reversed in the CAPE group, suggesting that the CAPE treatment significantly alleviated the disorder of the complement and coagulation cascades system in inflammatory zebrafishes. TNF-α has been reported to induce the expression of tissue factor (TF, or CD142), which activates factor X (F10) and initiates the extrinsic coagulation pathway [27,28]. In addition, cytokines promote platelet activation and aggregation, and increase the synthesis of coagulation factors such as fibrinogen in the liver [27,28]. Inflammation signaling damages vascular endothelial cells, which activates coagulation factor XII (F8) and initiates the intrinsic coagulation pathway [27,28].

### 3.3. CAPE Attenuates Pancreatic Exocrine Secretion and Inflammation Through CCKAR Modulation

It was reported that alcohol enhances the sensitivity of acinar cells of the pancreas to the CCK peptide and stimulates the release of CCK from duodenal I-cells [18,19]. CAPE might alleviate alcohol-induced inflammation due to its potential as an antagonist of CCK receptors, inhibiting the CCK-mediated pancreatic secretion. To explore the targeting proteins of CAPE in regulating pancreatic secretion pathway in alcohol-induced inflammatory zebrafishes, molecular docking was performed using the CAPE ligand and the key CCK receptor protein CCKAR.

The top three docking conformations between CAPE and CCKAR are shown in Figure 3b, with binding energies of −5.96, −5.71, and −5.36 kcal/mol (−24.93, −23.89, and −22.43 kJ/mol), respectively, among them. For the first docking mode, the phenolic hydroxyl group of CAPE molecules formed hydrogen bonds with Asn102 and Arg197, and the carbonyl group formed hydrogen bonds with Arg336. In addition, the phenethyl group engaged in hydrophobic interactions with Al332, Ala343, Leu347, and Ile352, while the benzene ring interacted with Met95, Leu99, Phe97, Pro101, and Phe107 with hydrophobic force. In the second mode, CAPE formed hydrogen bonds with Pro101, Asn102, Arg197, and Arg336, and had hydrophobic interactions with Phe107, Met195, Ala332, Ala343, Ile352, and Leu347. As for the third mode, CAPE formed hydrogen-bond interactions with Arg197 and Asn333, and had hydrophobic interactions with Pro101, Phe107, Met195, Ile329, Ala332, Leu347, Ile352, and Leu356. Despite slight differences among these molecular docking modes, CAPE was highly proximal to the reported substrate-binding sites of the CCK1 receptor (CCKAR) [29], which supported that CAPE could influence pancreatic secretion and related pathways by targeting the CCKAR protein.

Through the targeting of CCKAR, CAPE might inhibit the Gq protein and PKC signaling to inhibit zymogen granule secretion in the pancreatic secretion pathway based on the KEGG analysis transcriptomic data (Figure 3a). Thus, CAPE might inhibit the fusion of zymogen granules and the release of pancreatic amylases, lipases, and protease, ultimately achieving a down-regulation of digestive enzyme synthesis and secretion. In addition, CAPE improved the ER stress through inhibiting the influx of Ca^2+^ to ER mediated by Ca^2+^ pumps of SERCA (Sarco/Endoplasmic Reticulum Ca^2+^ ATPase), and increasing the Ca^2+^ efflux from ER mediated by the activated IP3R receptor to the fusion pathway of zymogen granules, and meanwhile increasing the Ca^2+^ secretion to pancreatic juice mediated by the CaCC (Ca-activated Cl channel) protein. Our study suggested that CAPE could significantly regulate pancreatic secretion by potentially targeting CCKAR, and could possibly be associated with alleviated ER stress and inflammation in alcohol-induced zebrafish.

### 3.4. Effect of CAPE on Gut Microbiota Associated with Inflammation in Zebrafish

The impact of CAPE treatment on the gut microbiota in the alcohol-induced zebrafishes was investigated. Compared to the control group, the α-diversities of the microbial community were influenced in the EtOH group and the CAPE group. In the EtOH and CAPE groups, the Ace, Chao, and Sobs indices based on ASV level showed higher values than those in the control group (Figure 4a). Significant differences were observed in the Ace, Chao, and Sobs indices between the CAPE group and the control group (*p* < 0.05, *p* < 0.05, and *p* < 0.05, respectively). Moreover, no significant differences were found in the Shannon and Simpson evenness indices among the three groups (Figure 4b). As indicated by the microbial dysbiosis index (MDI) (Figure 4c), the intestinal microbiota was significantly disordered in the EtOH group compared to the control group (*p* < 0.01). The Non-metric Multidimensional Scaling (NMDS) analysis and PCoA based on ASV levels confirmed the differences in the compositions of the microbial community among these groups (Figure 4d,e). To sum up, CAPE treatment could reshape the structures of the intestinal flora, which might associate with its alleviative effect on the intestinal microbiota-related inflammation in alcohol-induced zebrafishes.

The abundances of intestinal communities were analyzed, and the result showed that the microbial compositions changed in phyla, families, and genera levels among the three groups (Figure 5a). Among them, the phylum Proteobacteria was the most abundant, accounting for 93.8%, 81.0%, and 93.7% of the intestinal community in the control group, EtOH group, and CAPE group, respectively (Figure 5b). In addition, the abundance of phylum Bacteroidota increased significantly in the EtOH group compared to the control group (*p* < 0.01), but CAPE treatment decreased its abundant percentage to 4.99%. Similarly, the intestinal community in class level showed significant alteration among the three groups (Figure 5c). The linear discriminant analysis (LDA) effect size (LEfSe) based on ASV levels was used to identify significant different microbial communities between different groups. In the control group, order Fusobacteriales and genus *Cetobacterium* were the main discriminant communities. In contrast, phylum Bacteroidota, class Bacteroidota, Flavobacterial, and Alteromondales, family Weeksellaceae, Alteromondaceae, and genus *Flavobacterium*, *Chryseobacteria*, and *Rheinheimer* in the EtOH group showed LDA scores higher than 3.0 (Figure 5d). CAPE treatment increased the abundance of class Alphasproteobacteria, order Rhizobiales, family Rhizobiaceae, Rhodobacteraceae, and Pseudomonadaceae, and genus *Allorhizobium*, *Paracoccus*, and *Bosea*.

Correlation analysis indicated a tight association of many key genes in inflammation, lipid metabolism and sterol metabolism pathways with specific intestinal communities in alcohol-induced zebrafish (Figure 5e). The genus *Flavobacterium* was significantly positively correlated with *CEL1*, *CEL2*, and *LPIN* (*p* < 0.01), as well as *SOAT1*, *ALDH3A2B*, and *NPC1L11* (*p* < 0.05), which suggested that *Flavobacterium* was associated with the lipid metabolism disorder and inflammation. Genus *Mycoplasma* from Firmicutes was positively correlated with *PLPP2A*, *LPIN*, and *NPC1L1* (*p* < 0.05), and was extremely significantly positively correlated with *CYP24A1* (*p* < 0.01). The genus *Cetobacterium* was positively correlated with *MOGAT3A* and *DGAT2*, while the genus *Rhodococcus* from Actinobacteriota was negatively correlated with *MOGAT3A* and *LPL* (*p* < 0.05). In addition, the expressions of *CEL1*, *CEL2*, *SOAT1*, *ALDH3A2B*, *CYP24A1*, *TGL2*, *MOGAT3A*, *PLPP2A*, *LPL*, *DAGAT2*, and other genes were generously positively correlated with the abundance of genus *Pseudomonas*, *Perlucidibaca*, *Shewanella*, and *Acidovorax* from the phylum Proteobacteria, and negatively correlated with *Aeromonas*, *Xanthobacter*, and *Gemmobacter*. Therefore, CAPE could improve intestinal microbiota dysbiosis in alcohol-induced zebrafish, which was associated with the reduced inflammatory reaction and improved metabolic function.

## 4. Discussion

CAPE has been identified to be efficacious in ameliorating glucose and lipid metabolism disorders and inflammation. Our study indicated that CAPE significantly decreased the expressions of TNF-α and IL-1β and increased the expression of IL-10 in alcohol-induced inflammatory zebrafish. In addition, CAPE effectively regulated the pancreatic secretion, complement and coagulation cascades, protein digestion and absorption, and lipid digestion and absorption pathways according to KEGG enrichment analysis. In addition, molecular docking supported the potential of CAPE in targeting CCKAR and mediating the regulation of pancreatic secretion and inflammatory pathways. In recent years, various intestinal peptide hormones have been found to directly or indirectly influence pancreatic endocrine function, digestion and absorption, immunity, and metabolic function, including secretin, glucagon-releasing peptide, gastrin, somatostatin, and leptin and ghrelin [30]. This study supported that CAPE might impact intestinal function, inflammation, and immunity in alcohol-induced inflammatory zebrafish through targeting pancreatic secretion in an intestine–pancreas axis.

CAPE might serve as a CCKAR antagonist and improve pancreatic secretion disorder and inflammation. According to the molecular docking results, CAPE may bind to CCKAR in multiple ways (Figure 3b), and the binding site is proximate to the reported CCKAR active sites. It was reported that CCKAR interacts with the N-terminal dipeptide of the CCK8 through Trp39 and Gln40, and with the Met residue of the CCK8 through Leu50, Ile51, Leu53, Cys94, Met121, Ile351, Leu356, and Tyr360. In addition, Val125, Phe242, Trp326, Ile329, and Phe330 may constitute the binding site for the Phe residue of CCK8 [31]. Zhao et al. found that the mutation of Lys105 and Arg197 to Leu and Met of CCKAR, respectively, led to a substantial reduction in the receptor’s affinity for the CCK8 [32]. Arg197 of the CCKAR was believed to be the binding site for the sulfonic acid group of the CCK8, while Asn333 and Arg336 might be the binding sites for the C-terminal amino acids of the CCK8. According to the molecular docking in this study, CAPE could form hydrogen-bond interactions with the amino acid residues Asn102, Arg197, and Arg336 of the CCK receptor, and hydrophobic interactions with Al332, Ala343, Leu347, Ile352, Met95, Leu99, Phe97, Pro101, and Phe107. Therefore, CAPE could alter the conformation of CCKAR through at least partially blocking its binding sites, and potentially inhibiting the activation of the CCKAR-Gq-PLC pathway and the excessive pancreatic secretion induced by ethanol [33].

The suppression of CCKAR-mediated pancreatic secretion would influence the digestive zymogens and intestinal absorption function. We found that the expression levels of various digestive enzymes (Figure 3a) were markedly down-regulated in the CCKAR-mediated pancreatic secretion pathway, which might alleviate the disorder of the food digestion, gut barrier, microbiota, and microenvironment in alcohol-induced zebrafish. CCK-8 was reported to alleviate LPS-induced IL-1β increase in macrophages mainly by modulating PKA, p38, and the NF-κB pathway [34]. The CCK receptor was also reported to activate the mitogen-activated protein kinase (MAPK) signaling pathway, thereby influencing cell growth and differentiation [35]. These findings supported that CAPE might alleviate inflammation and health based on targeting the CCKAR in alcohol-induced zebrafish through multiple signaling pathways.

CAPE mitigated the onset and progression of the inflammatory response by modulating lipid metabolism and other signaling pathways. In this study, further analysis based on KEGG enrichment analysis using transcriptomic data revealed that CAPE regulated lipid metabolism and signal transduction, steroid biosynthesis, and fat digestion and absorption. In addition, CAPE treatment alleviated the increased gene expression in glycolipid metabolism and steroid biosynthesis pathways in hepatocytes of alcohol-induced zebrafish. CAPE treatment inhibited the generation of fatty acids from 1,2-diacyl-sn-glycerol and triacylglycerol by decreasing the levels of *TGL2* and *CEL*, and subsequently inhibited the fatty acid degradation. The expression levels of *LPL*, *PLP1*_2_3, *LPIN*, *MGAT2*, *DGAT2*, and *DGKE* were also down-regulated 2.5, 20.4, 3.7, 4.9, 8.1, and 2.5 times, respectively (Table 1), which confirmed that the significantly upregulated glycerophospholipid metabolism in alcohol-induced zebrafishes was reversed by CAPE treatment. Numerous studies have elucidated that inflammatory signals rapidly reprogram the lipid metabolic pathways of immune cells [36,37]. This reprogramming involves the activation of specific lipid-modifying enzymes and transporters. A recent investigation revealed that the genetic deletion of ceramide synthase 2 (Cers2), the enzyme responsible for the synthesis of very-long-chain (VLC) ceramides, inhibited the exacerbated inflammatory response associated with IL-10 deficiency [38]. Based on the tight connection between the inflammatory reaction and lipid metabolism disorders [39], the improved lipid metabolism disorder in the CAPE group might also benefit in alleviating the alcohol-induced inflammation. It has been reported that the preventive effect of CAPE on inflammation and metabolic disease is also associated with its antioxidative capability [40]. Therefore, CAPE might also exert its anti-inflammatory effect due to the antioxidant activity in the alcohol-induced zebrafishes.

Dietary substances can directly or indirectly affect the gut microbiota, which in turn impacts host physiology [41,42]. It is reported that a 10% fat diet rich in saturated fatty acids induced fat accumulation in zebrafish, resulting in gut microbiota dysbiosis and inflammation in zebrafish larvae, indicating the correlation between gut microbiota and systemic inflammation [43]. In this study, CAPE ameliorated the gut microbiota disorder in alcohol-induced zebrafish. Correlation analysis showed the tight relation of gut microbiota to the gene expression of certain pathways based on the transcriptomic data. Species of the genus *Flavobacterium* exist in a variety of ecological niches, and many of them cause serious disease in different freshwater fish species [44]. It was observed that genus abundance of *Flavobacterium* from Bacteroidota was significantly positively correlated with fatty acid and glyceride metabolism-related genes *CEL1*, *CEL2*, and *LPIN* (*p* < 0.01 for all), and with the expression of *SOAT1*, *ALDH3A2B*, *NPC1L1*, etc. (*p* < 0.05), indicating its important potential roles in regulating metabolic and inflammatory pathways in the alcohol-induced zebrafishes. Moreover, CAPE decreased the high abundance of Mycoplasmataceae induced by alcohol and improved gut microbiota dysbiosis. Mycoplasma is regarded as a pathogen that causes chronic infections [45]. Phylum Fusobacteriota and genus *Cetobacterium* have been reported to regulate insulin secretion and glucose consumption and metabolism. They were significantly increased in the EtOH group and decreased in the CAPE group, which suggested that excessive *Cetobacterium* might result in a disorder of gut microbiota and metabolic dysfunction. Therefore, CAPE might alleviate gut microbiota dysbiosis and intestinal function in zebrafish through inhibiting pathogenic bacteria, which was tightly associated with the alteration in inflammation, lipid metabolism, and intestinal health.

In addition, the activation of the CCKAR also stimulates the gallbladder contraction for bile release, and bile acids (BAs) in turn regulate physiological responses through the interaction with CCKAR [46]. CAPE might improve intestinal function and inflammation through regulating key metabolites of gut microbes. BAs are synthesized from cholesterol in the liver and then secreted into the intestine via the bile ducts, which play a pivotal role in regulating lipid metabolism, intestinal microbiota, and inflammation [24,47]. Gut microbes can regulate the physiological function of the host by transforming primary BAs to secondary BAs. In our study, the bile secretion and primary BA synthesis pathway were inhibited in the CAPE group compared to the control group. Therefore, the gut microbiota improved by CAPE treatment might alleviate inflammation and metabolic dysfunction through multiple signaling pathways.

## 5. Conclusions

CAPE exhibits significant anti-inflammatory effects in alcohol-induced inflammatory zebrafish. The results of transcriptomic analysis in this study indicated that the anti-inflammatory effect of CAPE is closely associated with the pancreatic secretion and the digestive pathways of lipid and proteins. Molecular docking supported the potential of CAPE in targeting CCKAR and mediating the regulation of pancreatic secretion. In addition, the improved gut microbiota in the CAPE group was also identified to be associated with its ameliorative effect in alcohol-induced inflammatory zebrafishes. This study emphasized the anti-inflammatory mechanisms of CAPE on its targeting of pancreatic secretion function, which will broaden the application of natural antioxidantds in improving metabolic and inflammatory problems.

## Figures and Tables

**Figure 1 biomolecules-15-00918-f001:**
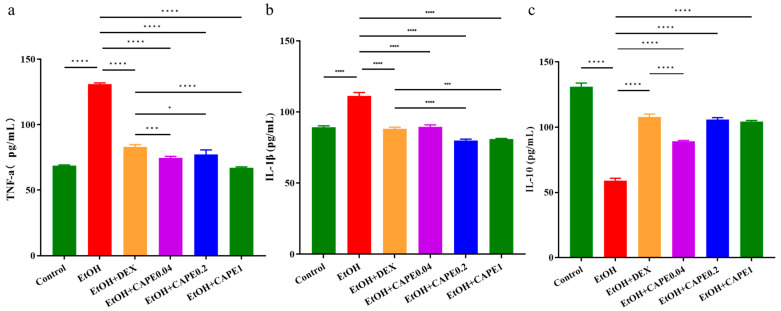
CAPE ameliorated alcohol-induced liver inflammation in zebrafish. The serum levels of TNF-α (**a**), IL-1β (**b**), and IL-10 (**c**). Control, EtOH (alcohol-induced group), EtOH + DEX (positive control with dexamethasone), EtOH + CAPE 0.04, EtOH + CAPE 0.2, and EtOH + CAPE 1 (0.75% ethanol model treated with 0.04 μg/mL, 0.2 μg/mL, and 1 μg/mL CAPE, respectively). Data are presented as mean ± SD (n = 6 for a, b, and c). *, *p* < 0.05; ***, *p* < 0.001, and ****, *p* < 0.0001 (One-way analysis of variance and Tukey’s multiple comparison test).

**Figure 2 biomolecules-15-00918-f002:**
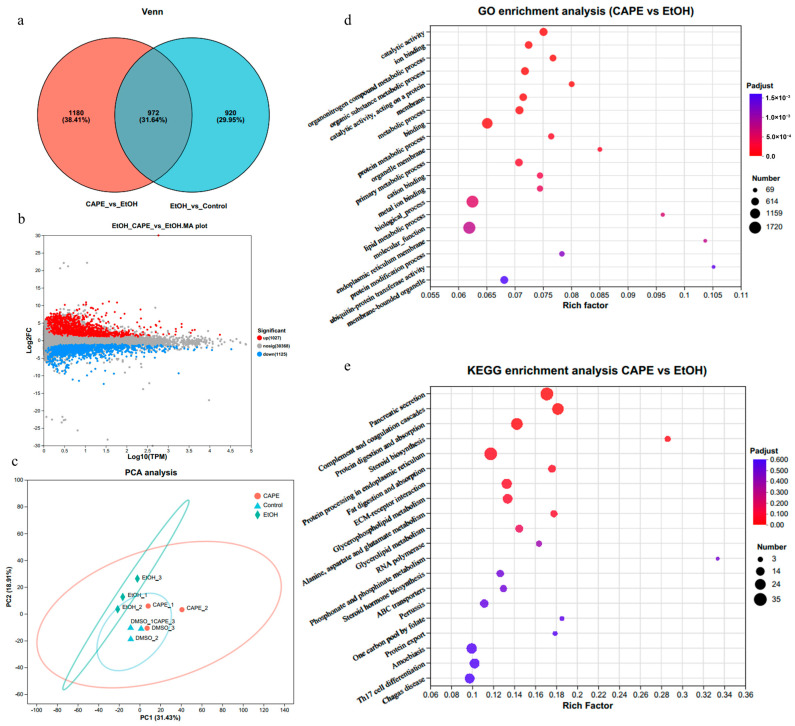
Effects of cape on the transcriptomic changes in the alcohol-induced inflammatory zebrafish. (**a**) Venn diagram of differentially expressed genes. (**b**) M-versus-A plot of differential expression levels. The values on both the horizontal and vertical axes have been logarithmized. The DESeq2 software 1.24.0 for differential gene analysis was used, and the significance-for-screening threshold was set at 0.05, with the Benjamini–Hochberg (BH) method for multiple test correction. (**c**) The principal component analysis (PCA) of transcriptomic levels. (PC1 and PC2 accounted for 31.43% and 18.91%, respectively). (**d**) GO enrichment analysis. (**e**) KEGG enrichment analysis. The horizontal axis represents GO or KEGG pathways, and the vertical axis represents the Rich factor.

**Figure 3 biomolecules-15-00918-f003:**
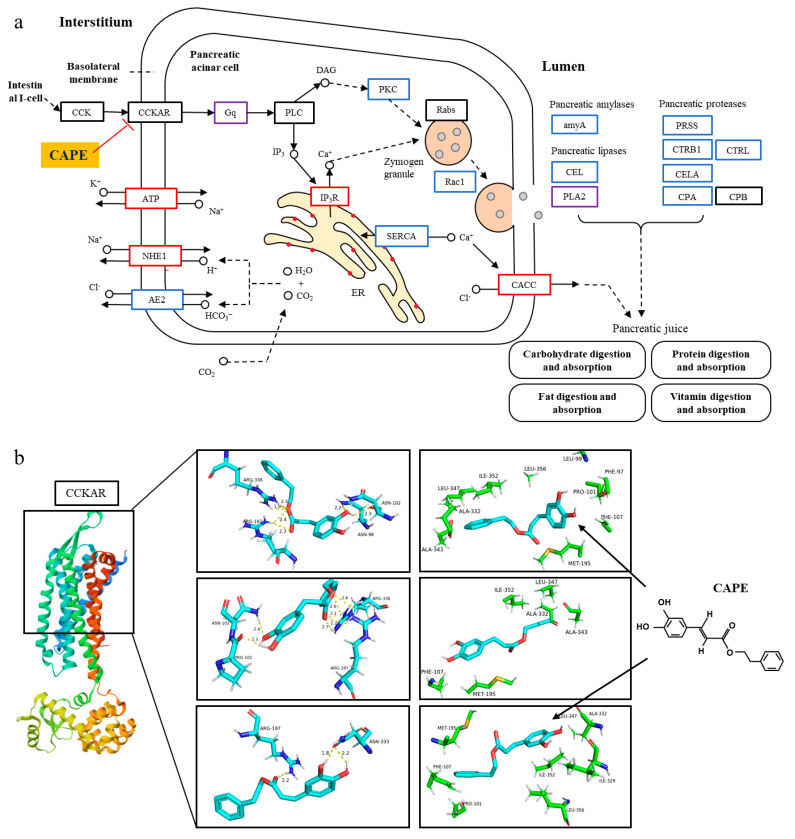
The potential mechanism of CAPE in regulating pancreatic secretion based on CCKAR. (**a**) The potential mechanism of CAPE in regulating the pancreatic secretion pathway. Red, blue, purple, and black boxes indicate genes with significantly upregulated expression levels, downregulated levels, both upregulated and downregulated levels, and no significant change, respectively. (**b**) Molecular docking was used to explore the potential binding conformations between CAPE and CCKAR. CAPE and the CCKAR protein interact via hydrogen bonds and hydrophobic interactions. The binding energy of docking manners increases from top to bottom. Light blue, white, dark blue, and red represent carbon (C), hydrogen (H), nitrogen (N), oxygen (O), and sulfur (S) atoms, respectively. The numbers indicate the interatomic distances, with the unit being angstroms (Å). Amino acid residues within five angstroms near the ligand CAPE are shown here. The structure on the far right is the molecular structure of CAPE.

**Figure 4 biomolecules-15-00918-f004:**
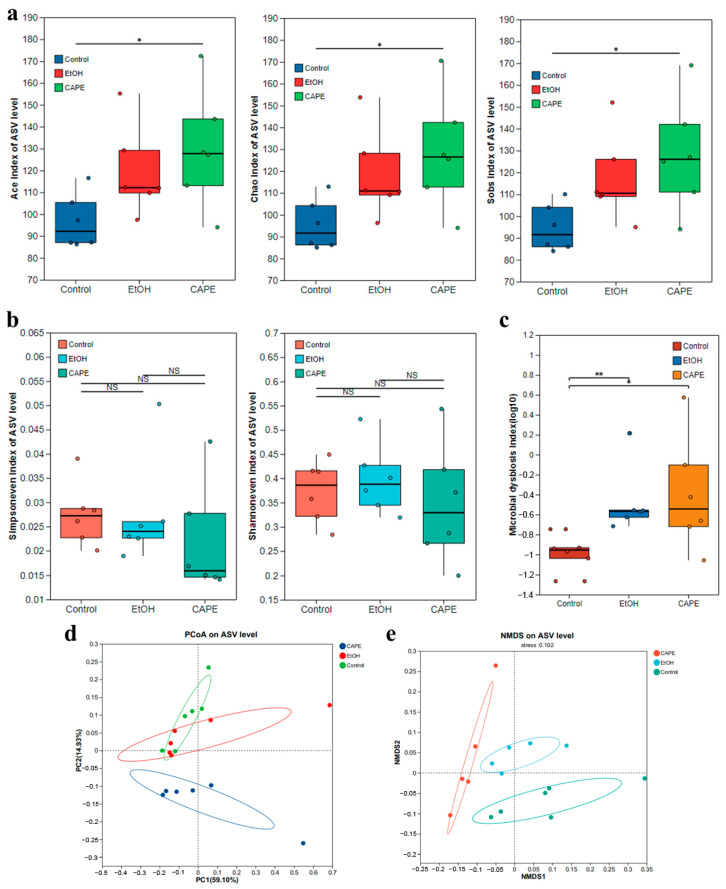
CAPE affects the composition of the intestinal microbiota of zebrafish induced by alcohol. (**a**) The Sobs, Chao, and Ace indices of α-diversity, which reflect the richness of the community. Analyses were conducted using the boot (1.3.18) and stats (3.3.1) packages in R (version 3.3.1). *, *p* < 0.05. (**b**) The Shannon and Simpson evenness indices reflect the evenness of the community. (**c**) The MDI. “**” indicates that the *p*-value is less than 0.05 according to the Wilcoxon rank sum test. (**d**) Principal component analysis (PCA). The horizontal and vertical coordinates represent the contribution values of the principal coordinates to the compositional differences in the samples. Different colors represent different groups, and the Bray–Curtis distance algorithm was used. (**e**) Non-metric Multidimensional Scaling (NMDS) analysis. Points in different colors or shapes represent samples from different groups. The closer two sample points are, the more similar the species composition of the two samples is.

**Figure 5 biomolecules-15-00918-f005:**
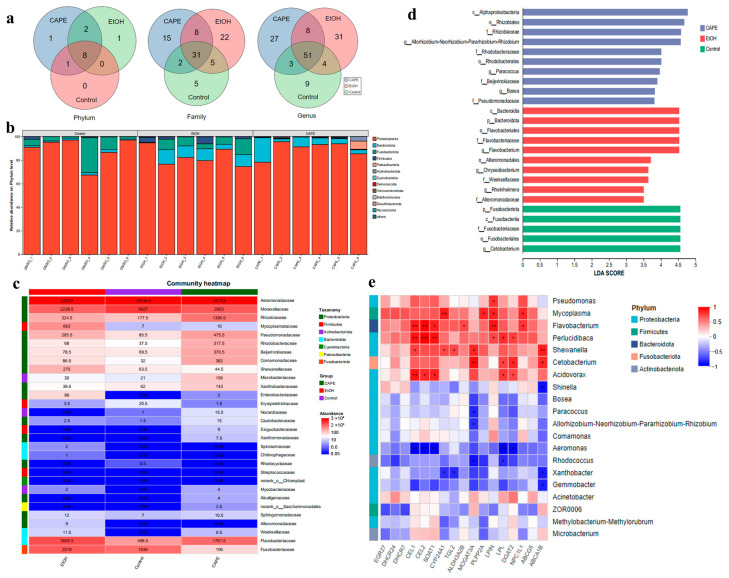
Changes in the composition of zebrafish gut microbiota induced by alcohol under the influence of CAPE. (**a**) Venn plots of relative abundance of gut microbiota at the phylum, family, and genus levels in the control, EtOH, and CAPE groups. (**b**) Bar plot of the community at the phylum taxonomic level (n = 6). (**c**) The distribution of dominant species at the family level in different groups (top 30), and the color gradient indicates the proportion of species. (**d**) Linear discriminant analysis (LDA) effect. The bar graph displays the LDA values of different differentially expressed species, showing species with LDA > 3 from the phylum level to the genus level. A one-gain-all (less strict) multi-group comparison strategy is used to analyze the absolute abundance of species. (**e**) Correlation analysis of lipid metabolism- and sterol metabolism-related genes and intestinal microbes in zebrafish. The top 20 genera in abundance were used for Spearman correlation analysis. The R value is displayed in different colors in the figure, and the species and clinical factor clustering tree is presented on the left and upper sides. *, *p* < 0.05; **, *p* < 0.01, ***, *p* < 0.001.

**Table 1 biomolecules-15-00918-t001:** CAPE affects the expression of differentially expressed genes in zebrafish.

KEGG Pathways	Gene Name	Control	EtOH	CAPE	FC (EtOH/CAPE)	*p* _adjust_
Coagulation cascade	*F9*	34.48	118.63	28.40	4.18	0.020
	*F8*	0.56	3.42	0.28	12.37	0.000
	*F10*	4.01	7.72	4.14	1.87	0.009
	*F5*	137.91	191.25	115.55	1.66	0.020
	*F2*	755.30	1130.58	464.18	2.44	0.028
Complement cascade	*C3*	302.59	629.66	141.02	4.46	0.000
	*C5*	52.43	232.52	36.94	6.29	0.002
	*C7*	246.62	453.23	23.85	19.00	0.001
	*C8A*	172.93	329.74	140.05	2.35	0.013
	*C9*	1290.50	3452.56	706.43	4.89	0.001
Steroid biosynthesis	*EGR27*	11.53	16.97	5.88	3.61	0.031
	*DHCR24*	6.13	22.80	2.52	9.80	0.000
	*DHCR7*	8.05	16.53	5.96	2.40	0.045
	*CEL1*	13.92	41.15	11.78	4.93	0.000
	*CEL2*	17.19	84.36	22.93	5.75	0.002
	*SOAT1*	5.68	14.24	5.30	2.92	0.028
	*CYP24A1*	26.31	248.35	14.38	23.81	0.000
Glycerolipid metabolism	*TGL2*	112.44	239.34	115.32	2.81	0.024
	*ALDH3A2B*	1.88	14.67	1.47	12.50	0.000
	*MOGAT3A*	40.47	61.94	17.98	4.93	0.011
	*PLPP2A*	0.32	2.80	0.17	20.41	0.000
	*LPIN*	0.27	1.18	0.43	3.73	0.030
	*LPL*	81.40	124.00	65.99	2.56	0.014
	*DGAT2*	13.58	27.54	4.58	8.06	0.000
Fat digestion and absorption	*NPC1L1*	4.06	7.33	4.15	3.61	0.004
	*ABCG5*	5.14	10.31	4.27	3.05	0.019
	*ABCA1B*	2.13	6.61	1.44	8.40	0.011

Note: The control group received normal feeding, and the EtOH group was the alcohol-induced group. The concentration of the CAPE group was 1.0 mg/L. FC represents the fold change, calculated as the gene expression level of the EtOH group divided by that of the CAPE group. *p*_adjust_ is the value after BH multiple correction for the comparison between the CAPE and EtOH groups. *F9* represents the coagulation factor IX gene, and *C3* represents the complement component 3 gene, etc.

## Data Availability

The original contributions presented in this study are included in the article. Further inquiries can be directed to the corresponding author.

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
