# Peer review of "Caffeic Acid Phenethyl Ester Alleviates Alcohol-Induced Inflammation Associated with Pancreatic Secretion and Gut Microbiota in Zebrafish"

_biomolecules, 2025, doi:10.3390/biom15070918_

Round 1
Reviewer 1 Report
Comments and Suggestions for Authors
In this study, the authors investigate the anti-inflammatory effects of caffeic acid phenethyl ester (CAPE) using a zebrafish model of alcohol-induced liver inflammation. The research combines in vivo experimentation, transcriptomic analysis, molecular docking, and gut microbiota profiling to elucidate the molecular mechanisms underlying CAPE’s effects. This work represents a valuable contribution to the understanding of natural bioactive compounds in the context of inflammation and metabolism, and it opens the door for further development of CAPE as a potential therapeutic agent.
Minor Comments and Suggestions:
L54: Replace the incorrect phrase “alcoholic-induced” with “alcohol-induced”. The same error appears again in L68.
L57: Correct the grammatical structure. It should read: “Zebrafish are characterized…”, using the plural form in scientific contexts.
L71: Replace “is identified” with the correct form: “has been identified” to maintain proper verb tense.
L82: Grammar issue: “Our previous study have revealed” should be corrected to “Our previous study has revealed…”.
The Materials and Methods section is clearly and concisely written. However, I suggest the authors include a schematic diagram of the experimental design, summarizing the treatment groups, sampling, and analyses performed. This would significantly improve clarity text for readers.
L132–136: Since you included a positive control group, I recommend identifying the group with “normal fish-rearing water” as the negative control for clarity and consistency in experimental terminology.
L150: Improve the wording by replacing “via MJZol total RNA extraction kit…” with “mRNA was isolated using the MJZol kit…”.
L222: Avoid starting the sentence with “And.” A more appropriate transition would be: “Moreover, the regulatory effects of CAPE…”
L257–263: It is advisable to provide full terms followed by abbreviations in parentheses the first time they appear in the text, e.g., Biological Process (BP), Cellular Component (CC), etc.
Reviewer 2 Report
Comments and Suggestions for Authors
Manuscript “Caffeic acid phenethyl ester alleviates alcohol-induced inflammation in zebrafish associated with pancreatic secretion and gut microbiota” describes the study of caffeic acid phenethyl ester (CAPE). This is a bioactive compound from propolis that has shown effectiveness in reducing liver inflammation and improving glucose and lipid metabolism disorders in an alcohol-induced zebrafish model. The study revealed that CAPE treatment decreased pro-inflammatory markers while enhancing anti-inflammatory responses, and its effects were linked to the regulation of pancreatic secretion and improvements in intestinal microbiota.
Modeling processes on zebra fish is a current trend in modern medicine. Manuscript is based on 48 literature references most of which are recent.
Thus this manuscript could be accepted for publication in the Biomolecules journal after addressing the following issue:
- Experimental section needs additional information about the CAPE used in your study. Was it purchased or synthesized? If it was purchases please add supplier name and quality of the product. If it was synthesized provide description of the process and add scheme of synthesis.
Reviewer 3 Report
Comments and Suggestions for Authors
The manuscript is well done. The authors employ various methods to demonstrate the anti-inflammatory effects of Caffeic acid phenethyl ester (CAPE) in the intestinal tract. However, they could significantly improve their manuscript by incorporating morphological techniques. Additionally, the concurrent use of antibodies against inflammatory molecules, along with western blot analysis and immunohistochemistry, greatly enhances the scientific weight of their work.
Comments on the Quality of English LanguageThe English should be evaluated by a mother language English reader.
Round 2
Reviewer 3 Report
Comments and Suggestions for Authors
The authors substantially didn't address my concerns. However, the last decision regards the Editorial Board of Biomolecules.